# Language Model is Suitable for Correction of Handwritten Mathematical Expressions Recognition

**Zui Chen**[1,2]**, Jiaqi Han**[1,2]**, Chaofan Yang**[1]**, Yi Zhou**[3]

[1]Shanghaitech University,
[2]Shanghai Innovation Center for Processor Technologies
[3]University of Science and Technology of China
[1]{chenzui, hanjq2022, yangchf}@shanghaitech.edu.cn
[3]yi_zhou@ustc.edu.cn

## Abstract

Handwritten mathematical expression recognition (HMER) is a multidisciplinary task that generates LaTeX sequences from images. Existing approaches, employing tree decoders within attention-based encoder-decoder architectures, aim to capture the hierarchical tree structure, but are limited by CFGs and pre-generated triplet data, hindering expandability and neglecting visual ambiguity challenges. This article investigates the distinctive language characteristics of LaTeX mathematical expressions, revealing two key observations: 1) the presence of explicit structural symbols, and 2) the treatment of symbols as minimal units, each directly assigned specific semantics. Rooted in these properties, we propose that language models have the potential to synchronously and complementarily provide both structural and semantic information, making them suitable for correction of HMER. To validate our proposition, we propose an architecture called **R**ecognition and **L**anguage **F**usion **N**etwork (RLFN), which integrates recognition and language features to output corrected sequences while jointly optimizing with a string decoder recognition model. Experiments show that RLFN outperforms existing state-of-the-art methods on the CROHME 2014/2016/2019 datasets.[1]

## 1 Introduction

Handwritten Mathematical Expression Recognition (HMER), a demanding subsection of optical character recognition (OCR), constitutes an interdisciplinary crossroad of computer vision, pattern recognition, and natural language processing (NLP). The unfolding of deep learning advancements has notably enhanced the effectiveness of HMER, ushering its adoption in diverse arenas, including intelligent education. Nonetheless, the precision of these technologies is continuously challenged by inherent ambiguities in handwritten characters and the complexity of mathematical formulas. These hurdles underscore the pivotal role that NLP could play in enhancing the robustness of current visual models grappling with these issues.

The encoder-decoder architecture is the prevalent method for HMER, which recasts the problem as an image-to-sequence translation task, converting a handwritten formula image into a LaTeX markup sequence. In contrast to traditional OCR tasks, the two-dimensional structure of handwritten formulas necessitates an approach that doesn't rely on direct segmentation. Since Zhang et al. (2017) introduces a decoder using RNN with attention, subsequent work has concentrated on enhancing the accuracy of the visual attention (Zhao et al., 2021; Bian et al., 2022; Li et al., 2022). Currently, various tree decoders and methods of syntactic analysis, such as Zhang et al. (2020) and Yuan et al. (2022), are employed to focus on analyzing the expression structure and the relations of symbols.

While structure-focused methods have undeniably enriched recognition model capabilities, they have also precipitated two notable challenges: 1) They rely on complex Context-Free Grammars (CFGs), necessitating the pre-transformation of the LaTeX markup sequence into specific tuple representations, which limits their extensibility. 2) The issue of visual ambiguity is left behind. Tree decoders pay less attention to context when predicting triples, often unable to distinguish differences such as '2' and 'z'. Ung et al. (2021) try to employ a language model (LM) for post-correction, but Gupta et al. (2021) underline the inherent risk of wholly depending on a LM for the correction of low-redundancy information, such as numbers, which is particularly susceptible to biases introduced by probabilistic skewing.

However, as a formal language designed for mathematical structures, LaTeX mathematical expressions possess unique language characteristics.

---

[1]https://github.com/Zui-C/RLFN

We believe that advance structural analysis separately following normal NLP methods may not be a prerequisite to catch complicated structures of LaTeX mathematical expressions.

Two key characteristics of LaTeX mathematical expressions are: 1) They have explicit structural symbols. 2) Minimal units are symbols, and they are directly assigned specific semantics. Based on this, we propose that LMs can proffer both structural and semantic information, making them suitable for correction of HMER. An in-depth theoretical and statistical exploration of this perspective is articulated in Section 3.

Specifically, regarding the current limitations of structure-focused methods, we believe that: 1) The structural information can be represented by structural symbols' semantic with the context provided by mathematical notations. This circumvents the necessity for complex CFGs to generate triplet data. 2) Character ambiguity between different types can be rectified with contexts provided by structural symbols and mathematical notations.

We substantiate our propositions through experiments on HMER correction. Specifically, we deploy a math LM to rectify an HMER model reliant on unstructured-based methods, demonstrating the suitability of LM in addressing current limitations. Additionally, we argue against the sole reliance on LMs in a post-correction method. By leveraging information from the recognition model, we can constrict the correction space.

Finally, we propose our architecture called **R**ecognition and **L**anguage **F**usion **N**etwork (RLFN), which integrates recognition and language feature to output correct sequences and optimizes jointly with the recognition model. Experiments show that RLFN outperforms existing state-of-the-art methods and achieves expression recognition rates (ExpRate)s of 57.00/54.23/54.13% on the CROHME 2014/2016/2019 datasets.

## 2 Related works

### 2.1 HMER

Many traditional methods utilize specially designed grammars, including Chan and Yeung (2001) that employ definite clause grammar, MacLean and Labahn (2013) that propose a fuzzy relational grammar for handling ambiguous and non-linear inputs, Álvaro et al. (2014) that apply hidden Markov models to CFG, and Noya et al. (2021) that integrate hypergraph into CFG prediction. While the above

methods treat symbol recognition and structure analysis separately, several global methods aim to tackle them simultaneously. Awal et al. (2014) consider HMER as a simultaneous optimization problem encompassing expression recognition, symbol recognition, and structure analysis. Then Álvaro et al. (2016) further extend the methodology by incorporating a 2D-PCFG to integrate stochastic information from multiple sources.

Encoder-Decoder based methods are led by Deng et al. (2017) and Zhang et al. (2017). Based on CNN encoder and RNN decoder, Deng et al. (2017) design a coarse-to-fine process, while Zhang et al. (2017) design coverage attention to avoid over-parsing and under-parsing. Zhang et al. (2018) use DenseNet (Huang et al., 2017) as encoder and introduce a decoder with multi-scale attention. Wu et al. (2020) integrate left-to-right attention to simulate the progressive nature of human perception. Wang et al. (2019) use multi-modal attention aim to fully utilize both online and offline information. Zhao et al. (2021) replace RNN-based decoder with a bidirectionally trained transformer, leading to enhance global coverage and parallelization capabilities. Bian et al. (2022) apply mutual learning to enhance bidirectional learning and design a multi-scale coverage attention for longer expressions.

Several works focus on the tree structure of math expressions. Zhang et al. (2020) regarded the expression as a tree represented by triples that include parent, children, and relation; then designed a tree decoder to predict each triple. Based on this work, Zhong et al. (2022) expanded prediction of symbols into attribute prediction and position prediction, then purposed a transformer-based decoder to predict triples. Yuan et al. (2022) utilized grammar constrained attention to transform the whole image into a parse tree. Wu et al. (2022a) added thinking attention to tree decoder, assisted by pixel-level auxiliary loss to improve recognition of complex expressions. Wu et al. (2022b) designed a structural string representation, attempting to utilize both language model and tree structure. These structured representations are specifically designed, limiting their extensibility.

### 2.2 HMER & OCR Correction

Limited research are done on HMER correction. Chan and Yeung (2001) detect and correct errors based on grammar and heuristics rules. Ung et al. (2021) train and apply a language model for post-

correction tasks.

More correction works are done on OCR. Litman et al. (2020) repetitively correct recurrent block outputs by fusing it with visual features each step in training. Other works use LM to assist correction. Nguyen et al. (2020) use BERT for error detection, and then use neural machine translator to correct errors. Qiao et al. (2020) use the pre-trained FastText (Joulin et al., 2017) to supervise the generation of semantic features, fusing it with encoder visual features to capture global semantic information. Gupta et al. (2021) utilize perplexity of language model to choose output among multiple aligned models, and stress that correction of numbers requires extra reliable information source. Yasunaga et al. (2021) adopt unsupervised correction by comparing logits of LM output with local perturbations of the text. Fang et al. (2021, 2023) explicitly use built-in bidirectional LM to iteratively correct the output.

Several math LMs are pretrained jointly with text and LaTeX expressions, potentially beneficial for HMER. Novotný and Štefánik (2022) design MathBERTa based on RoBERTa (Liu et al., 2019), with a soft vector space model to capture the semantic similarity between tokens. Peng et al. (2021) is designed to improve the prediction of masked formula substructures extracted from the Operator Tree (OPT). Scarlatos and Lan (2023) conduct multiple modifications on the GPT-2 (Radford et al., 2019) model, resulting in MathGPT, which exhibits strong performance in generating mathematical expressions. Our method utilizes MathBERTa to provide auxiliary information for correction.

## 3 Why LM is Suitable for HMER Correction?

### 3.1 Theoretical Analysis

As a formal language designed specifically for the representation of complex mathematical symbols and formulas, symbols in LaTeX mathematical expressions can be broadly divided into four distinct categories: 1) Structural symbols _, ^, {, }, \{, \} 2) Mathematical notations (e.g., \frac, \sqrt, +, -), 3) Latin and Greek alphabets (e.g., A, a, $\alpha$), and 4) Numbers. The key differences from English can be summarized in the following two points:

1) The structural symbols in LaTeX mathematical expressions explicitly convey their structure, and certain mathematical notions serve to assist in this structural representation. This mechanism of structural representation shares fundamental simi-

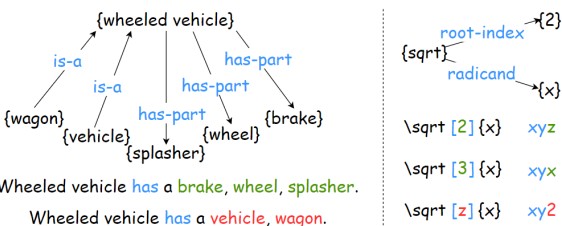

Figure 1: Cases of structural symbols explicitly convey their structure in the context of mathematical notations. Like in English, the same word has different semantics in different contexts.

Figure 2: Cases of the semantic problems within different types of symbols can be found with the structure and mathematical notation. Like in English, given the semantic relationship in WordNet, whether these types of words are correct or incorrect.

larities with the mechanism of how words within phrases in English explicitly communicate their semantics. And this characteristic enables math LMs to synchronously provide structural information just as they provide semantic information.

As figure 1 shows, '\frac' and '\sqrt' represent the fraction line and radical symbol itself while providing context. Then the following '{' respectively represent the beginning of the numerator and radicand. In the English case, 'parse' relies on different context to express different semantics in sentence 'parse a sentence' and 'parse words'.

2) LaTeX mathematical expressions treat symbols as minimal semantic units. And based on contextual semantics, visual ambiguities between categories can be corrected. Though using 'x' or 'y' as unknown variables has no difference, the semantic distinction between an unknown variable and a number is significant. For instance, $\sqrt[2]{x}$ is reasonable, but $\sqrt[z]{x}$ is not in line with convention. We will represent it as $x^{1/z}$ instead. Similarly, when we trust z, we tend to believe that the expression is not a radical expression.

From the perspective of analogy with English, Figure 2 illustrates why different category symbols in LaTeX mathematical expressions have semantic differences when given structural relationships. In English, sentence semantic errors

caused by certain types of words can be detected through semantic relationships. According to WordNet (Fellbaum, 2005) of 'wheeled vehicle', 'wagon' and 'vehicle' have an 'is-a' relation with it. While given 'has-part' relationship, semantic errors in sentence 'wheeled vehicle has a vehicle' can be discovered. Similarly, the symbol '[ ]' represents the 'root-index' in the case of '\sqrt[2]{x}' and '\sqrt[3]{x}'. '\sqrt[z]{x}' uses an unknown variable as the 'root-index' which is generally unconventional. Moreover, words that represent semantic relationships, such as 'multiply' and 'multiplied', are not explicitly stated in the case of 'xyz' and 'xyx'. In general, given the multiplicand (the left term of multiplication), it's expected to use an unknown number as the multiplier, or semantic errors may occur in the case of 'xy2'.

In addition to language characteristic that make LM suitable for HMER correction, the task itself is also suitable. While OCR employs letters as the smallest unit for correcting words, HMER utilizes symbols as the minimum unit for amending expressions. The former represents morphological correction, while the latter is semantic correction.

## 3.2 Statistical Analysis

We conducted statistical analysis as collateral evidence on the CROHME dataset (Mouchère et al., 2014). The CROHME dataset, a byproduct of the Competition on Recognition of Online Handwritten Mathematical Expressions (CROHME), is universally recognized as the principal public dataset within HMER field. A comprehensive collection, the CROHME training set comprises 8835 handwritten mathematical expressions. In addition, it includes three distinct testing subsets, namely CROHME 2014, 2016, and 2019, containing 986, 1147, and 1199 handwritten mathematical expressions respectively. Noteworthy is the inclusion of a total of 111 symbol classes, which encompasses the "sos" and "eos" symbols.

**The explicit structural symbols do express their semantics.** This is affirmed via an application of the Math-aware BERT model (Reusch et al., 2022), where the calculated perplexity acts as an index for semantic strength, applied to those four categories of symbols in CROHME training set.

In detail, we engage an individualized masking operation, followed by a model prediction of the obfuscated symbol. The outcome is a probability

| Type | SS | MN | LGA | Num | Total |
|---|---|---|---|---|---|
| **Perplexity** | 2.905 | 3.396 | 3.723 | 3.282 | 3.262 |
| **Counts** | 51030 | 35112 | 28275 | 22267 | 136684 |
| **Categories** | 6 | 39 | 54 | 10 | 109 |

Table 1: Perplexity, counts and categories of structural symbols (SS), mathematical notations (MN), latin and greek alphabets (LGA), and numbers (Num).

| Type | z↔2 | 9↔q | 0↔o | 9↔g | 5↔s | Top 5 |
|---|---|---|---|---|---|---|
| **s1→s2** | 1.926% | 2.627% | 1.226% | 1.576% | 0.876% | 8.231% |
| **s2→s1** | 1.226% | 0 | 1.226% | 0.350% | 0 | 2.803% |
| **Total** | 3.152% | 2.627% | 2.452% | 1.927% | 0.876% | 11.034% |

Table 2: Percentage of top 5 alphabet-number mis-recognition pairs among all SUB1 cases. $s1 \rightarrow s2$ indicates that symbol $s1$ is mis-recognized as $s2$.

distribution, the reciprocal of which, corresponding to the actual word, signifies its perplexity. Subsequently, the mean perplexity, according to category, is designated as the perplexity of this particular category of symbols.

Results in Table 1 reveal that among the four types of symbols, the structural symbols exhibit the lowest perplexity. This finding aligns with our theoretical analysis that the explicit structural symbols suggest a comparatively robust semantic signal.

**The visual ambiguity between numbers and alphabets do exist.** We conducted an analysis of the results from the string decoder recognition model, DWAP (Zhang et al., 2018), using the CROHME 2014/2016/2019 datasets. During this analysis, we employed a reliable metric called Substitute-by-One (SUB1), which identifies cases where the model's predictions deviate from the ground truth by only one substitution. Within the SUB1 cases, the mis-recognition of one character as another is verified, avoiding character substitution indeterminacy in evaluation.

The outcomes are shown in Table 2. Among all SUB1 instances, the top 5 pairs of mis-recognition between numbers and alphabets contribute to 11% of the total mis-recognition, while the overall mis-recognition between numbers and alphabets contribute to 26%. These statistical findings highlight that mis-recognition between numbers and letters not only exists considerably but also tends to concentrate on visual ambiguity. Thus overcoming the visual ambiguity issues as mentioned in theoretical analysis is significant.

## 4 Method

In an endeavor to empirically corroborate our hypotheses, we propose a novel architecture, the **R**ecognition and **L**anguage **F**usion **N**etwork (RLFN), engineered specifically to address the dual challenge of visual ambiguity and the complex structural issues that are inherent to string decoder recognition models. The RLFN is built with a string decoder recognition module, a language module that extracts language information, and a fusion module to refine the recognition output by utilizing the language information.

### 4.1 Recognition Module

Our recognition module basically follows DWAP (Zhang et al., 2018), using DenseNet (Huang et al., 2017) to extract visual feature $F \in \mathbb{R}^{H' \times W' \times D}$ from the single-channel input image.

As shown in Figure 3, each step $t$ in the decoder, we iteratively update two state weights: the GRU (Cho et al., 2014) hidden state $h_t$ and the coverage attention (cumulative attention map) $A_t$.

$$h_t = \text{GRU}(E\gamma_{t-1}, h_{t-1}) \tag{1}$$

$$e_t = W_e \tanh(F_p + W_A A_{t-1} + W_h h_{t-1}) \tag{2}$$

$$\alpha_{i,j;t} = \frac{\exp(e_{i,j;t} - \max_{i,j}(e_{i,j;t}))}{\sum\limits_{i,j} \exp(e_{i,j;t} - \max_{i,j}(e_{i,j;t}))} \tag{3}$$

$$A_t = A_{t-1} + \alpha_t \tag{4}$$

Here, $e_t$ represents the attention score which produces the attention map $\alpha_t$, where $i, j$ denote the coordinate on the feature map. $E\gamma_{t-1}$ represents the embedding of the last symbol, $F_p$ corresponds to the position encoded feature (Parmar et al., 2018). $W_e$, $W_A$, and $W_h$ represent the trainable weights. After that, we generate the content vector $c_t$ with element-wise multiplication of $\alpha_t$ with visual features $F$. Then $c_t$ is combined with $h_t$ and embedding of $\gamma_{t-1}$ to obtain the symbol state $s_t$ and the recognition prediction symbol $\gamma_t$.

$$s_t = W_c c_t + W_\gamma E\gamma_{t-1} + W_h' h_t \tag{5}$$

$$\gamma_t = \text{softmax}(w^\top s_t + b) \tag{6}$$

$W_c, W_\gamma, W_h', w, b$ are trainable weights. Lastly, recognition module outputs the total symbol states $s$ as recognition feature $F_R$, along with the recognition prediction sequence $\gamma$.

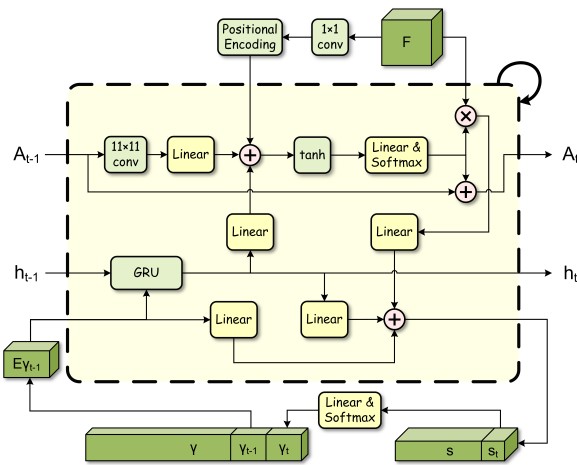

Figure 3: Recognition module

### 4.2 Language Module

We utilize MathBERTa (Novotný and Štefánik, 2022) to extract language information, which is a RoBERTa (Liu et al., 2019) model specifically fine-tuned on LaTeX expressions.

As a variant of BERT (Devlin et al., 2019), RoBERTa solely focuses on masked language modeling (MLM) task, uses larger mini-batches and employs dynamic masking, which all contribute to improve bidirectional semantic language modeling.

Building upon RoBERTa, MathBERTa further focuses on language processing of LaTeX expressions. It undergoes fine-tuning on an extensive dataset containing both text and LaTeX expressions. This specialized training enhances MathBERTa's comprehension of semantic and syntactical properties of LaTeX mathematical expressions.

Considering that our recognition output does not need tokenization, and LaTeX mathematical notations are prone to problems, we manually associate the vocabulary with the one-hot encoding in MathBERTa instead of using the tokenizer. Then given the recognition prediction sequence $\gamma$, MathBERTa outputs the language feature $F_L$.

### 4.3 RLFN

As shown in Figure 4, in RLFN, the input image is sent to recognition module to extract recognition feature and the prediction sequence. The latter is passed to language module to form language feature. Both features are fused in fusion module to output the corrected prediction sequence.

The main objective of the fusion module is to integrate the information from the recognition module with the semantic information obtained through

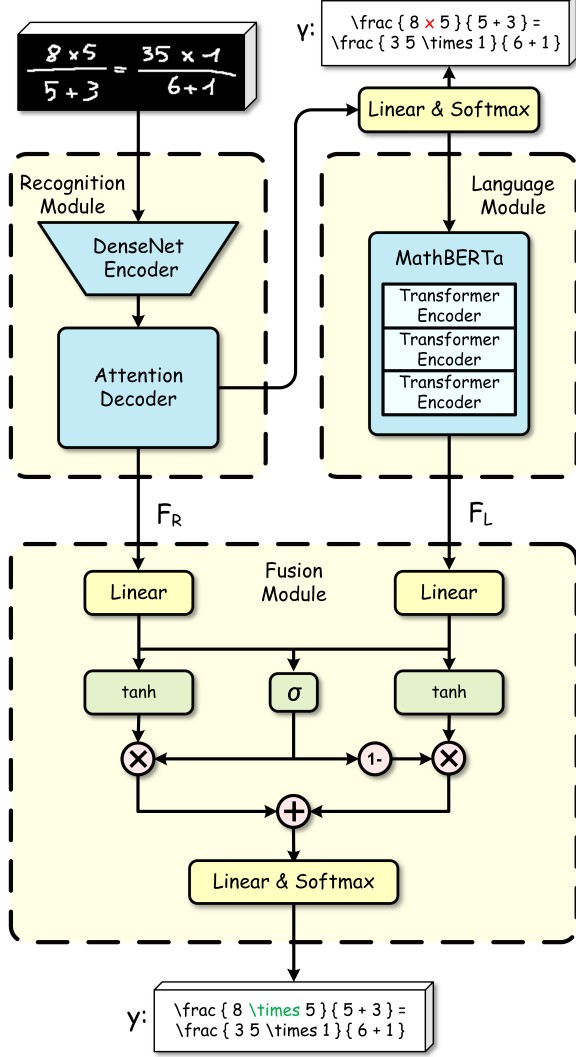

Figure 4: Architecture of **R**ecognition and **L**anguage **F**usion **N**etwork (RLFN)

the language module, so as to generate the corrected prediction sequence.

In the fusion module, the recognition feature $F_R$ and language feature $F_L$ are first dimensionally aligned through linear operation. Then we follow Yao et al. (2017) to incorporate a gating neuron, denoted as $\sigma$. This neuron allows us to assign weights based on contributions of two features during the calculation of output. Within the gating neuron, the two aligned features are horizontally concatenated. The resulting concatenated vector is then adjusted to match the size of the aligned features. Subsequently, the resized vector is fed into a sigmoid function to generate weights. These weights are then utilized to modulate the output of the aligned features, which are first processed through tanh activation function.

The process of generating the corrected sequence

$y$ in the fusion module is as follows:

$$x_R = W_R F_R, \quad x_L = W_L F_L \quad (7)$$

$$z = \sigma(W_F[x_R, x_L]) \quad (8)$$

$$h_R = \tanh(x_R), \quad h_L = \tanh(x_L) \quad (9)$$

$$h = z \cdot h_R + (1 - z) \cdot h_L \quad (10)$$

$$y = \mathrm{softmax}(w'^{\top} h + b') \quad (11)$$

where $\sigma$ refers to sigmoid activation function, $W_R, W_L, W_F, w', b'$ are weights to be learned.

### 4.4 Parameter Learning

We jointly optimize our RCFN with the recognition module through a linear classification layer, and the loss function is as follows:

$$\mathcal{L} = \mathcal{L}_R + \mathcal{L}_F \quad (12)$$

where $\mathcal{L}_R$ and $\mathcal{L}_F$ are the cross-entropy loss of the recognition prediction sequence probability and the corrected prediction sequence probability with respect to the ground-truth.

$\mathcal{L}_R$ aims to guide the the recognition module and the additional linear classification layer, while $\mathcal{L}_F$ focus on guiding the fusion of the two features. To mitigate the influence of the training set's probability bias and the large number of parameters, we have frozen the language model's parameters. Furthermore, gradient separation has been employed to enhance the focus of the loss functions on their respective optimization goals within the recognition model. Nevertheless, joint optimization still affects each other's updates of problematic parts through the optimizers and other means.

## 5 Experiments

### 5.1 Implement Details

Our RCFN is implemented in PyTorch with a single NVIDIA GeForce RTX 3090. We use Adadelta optimizer (Zeiler, 2012) with the learning rate increases from 0 to 1 at the first epoch and decays to 0 following the cosine schedules (Zhang et al., 2019b). No data augmentation for fair comparison. The batch size is set to 8. All images within a batch are filled in the upper left corner of the canvas of the same size. Due to memory limitations, the canvas size does not exceed 1280 * 280, and any excess images will be discarded. The total training epoch is set to 200 epochs taking around 16 hours.

| Method | CROHME 2014 | | | CROHME 2016 | | | CROHME 2019 | | |
|---|---|---|---|---|---|---|---|---|---|
| | ExpRate↑ | ≤ 1 ↑ | ≤ 2 ↑ | ExpRate↑ | ≤ 1 ↑ | ≤ 2 ↑ | ExpRate↑ | ≤ 1 ↑ | ≤ 2 ↑ |
| UPV (Mouchère et al., 2014) | 37.22 | 44.22 | 47.26 | - | - | - | - | - | - |
| WAP (Zhang et al., 2017) | 46.55 | 61.16 | 65.21 | 44.55 | 57.10 | 61.55 | - | - | - |
| PAL (Wu et al., 2019) | 39.66 | 56.80 | 65.11 | - | - | - | - | - | - |
| TAP (Zhang et al., 2019a) | 48.47 | 63.28 | 67.34 | 44.81 | 59.72 | 62.77 | - | - | - |
| DWAP (Zhang et al., 2018) | 50.10 | - | - | 47.50 | - | - | - | - | - |
| MAN (Wang et al., 2019) | 54.05 | 68.76 | 72.21 | 50.56 | 64.78 | 67.13 | - | - | - |
| PAL-V2 (Wu et al., 2020) | 48.88 | 64.50 | 69.78 | 49.61 | 64.08 | 70.27 | - | - | - |
| RBR (Truong et al., 2020) | 53.40 | 65.20 | 70.30 | 52.10 | 63.20 | 69.40 | 53.10 | 63.90 | 68.50 |
| DLA (Le, 2020) | 49.85 | - | - | 47.34 | - | - | - | - | - |
| DWAP-TD (Zhang et al., 2020) | 49.10 | 64.20 | 67.80 | 48.50 | 62.30 | 65.30 | 51.40 | 66.10 | 69.10 |
| WS-WAP (Truong et al., 2020) | 53.65 | - | - | 51.96 | 64.34 | 70.10 | - | - | - |
| BTTR (Zhao et al., 2021) | 53.96 | 66.02 | 70.28 | 52.31 | 63.90 | 68.61 | 52.96 | 65.97 | 69.14 |
| ABM (Bian et al., 2022) | 56.85 | **73.73** | **81.24** | 52.92 | 69.66 | **78.73** | 53.96 | 71.06 | 78.65 |
| SAN (Yuan et al., 2022) | 56.20 | 72.60 | 79.20 | 53.60 | 69.60 | 76.80 | 53.50 | 69.30 | 70.10 |
| GPT-4V (Yang et al., 2023) | 31.85 | 49.09 | 60.45 | - | - | - | - | - | - |
| DWAP (Baseline)† | 51.72 | 69.47 | 77.99 | 48.82 | 67.13 | 75.41 | 50.79 | 69.64 | 76.81 |
| RLFN-DWAP (Ours) | **57.00** | 72.01 | 80.73 | **54.23** | **70.10** | 78.47 | **54.13** | **72.56** | **80.07** |

Table 3: Results on the CROHME dataset without any data augmentation. †indicates that we reproduce DWAP as shown in figure 3. RLFN-DWAP represents we take the reproduced DWAP as our recognition module.

## 5.2 Evaluation

The metric of expression recognition rate (ExpRate) is utilized, defined as the proportion of accurately recognized expressions. Additional measurements, denoted as ≤ 1 and ≤ 2, are also employed, where the ExpRate accommodates at most one or two symbol-level errors, respectively.

We experiment on CROHME datasets mentioned in Section 3.1. Consistent with previous methods, we use CROHME 2014 as the validation set and test on CROHME 2016 and 2019 to compare with previous state-of-the-art (SOTA) methods.

As shown in Table 3, we take the reconstructed DWAP as our baseline. And our RLFN-DWAP using it as the recognition module achieves SOTA on the ExpRate indicator. In addition, it can be observed that the improvement of model on ExpRate is higher than on ≤ 1 or ≤ 2, which is consistent with the intuition that sentences with fewer errors have more complete semantics information.

Inspired by the LaTeX code generation capability reported in (Yang et al., 2023), we conduct an experiment using GPT-4V on CROHME 2014 dataset with the the prompt 'generate latex code and output without compile.' The outputs are post-processed to align CROHME vocabulary, and the ExpRate of GPT-4V is 31.85. Given that it is not finetuned on the CROHME dataset, its performance is acceptable.

## 5.3 Ablation Study

In this subsection, we perform a ablation study to analyze the impact of the language module and the

| Method | ExpRate↑ | ≤ 1 ↑ | ≤ 2 ↑ |
|---|---|---|---|
| DWAP (Baseline) | 51.72 | 69.47 | 77.99 |
| + Language Module | 53.96 | 70.18 | 78.80 |
| + Fusion Module | 57.00 | 72.01 | 80.73 |

Table 4: Ablation study on CROHME 2014

fusion module. To separate the impact of the language module, we did not use the fusion module in RLFN. Instead, in order to generate the corrected prediction sequence, we treat it as a translation task and add a decoder with two transformer layers in the language module. Other settings are all identical to RLFN. Results on CROHME 2014 are shown in table 4, we can only tell that the language module and the fusion module both have their impact.

## 5.4 Improvement Study

In this section, we explore whether the improvement comes from the correction of complex structure and visual ambiguity to validate our proposition that LM do obtain semantic and structural information synchronously and complementarily.

We conduct an incremental comparison on structural complexity to assess the improvement of our model on complex expressions across all datasets. We define the structural complexity of an expression as the count of six structural symbols mentioned in Section 3.1. The results are presented in Table 5. Models perform worse when facing more complex expressions, suggesting they are more challenging to recognize. Interestingly, the relative improvement becomes progressively higher for more complex expressions. The improvement

| | | DWAP | RLFN | RI |
|---|---|---|---|---|
| **Complex 25%** | ExpRate | 0.2965 | 0.3469 | 17.00% |
| | $\leq 1$ | 0.4454 | 0.4766 | 7.01% |
| | $\leq 2$ | 0.5378 | 0.5954 | 10.71% |
| | $\leq 3$ | 0.6074 | 0.6543 | 7.71% |
| **Complex 50%** | ExpRate | 0.3908 | 0.4322 | 10.60% |
| | $\leq 1$ | 0.5672 | 0.6014 | 6.03% |
| | $\leq 2$ | 0.6507 | 0.6963 | 7.01% |
| | $\leq 3$ | 0.7209 | 0.7539 | 4.58% |
| **Total** | ExpRate | 0.5039 | 0.5501 | 9.17% |
| | $\leq 1$ | 0.6873 | 0.7155 | 4.10% |
| | $\leq 2$ | 0.7668 | 0.7971 | 3.95% |
| | $\leq 3$ | 0.8235 | 0.8457 | 2.70% |

Table 5: Incremental examination of the Top 25%, 50%, and total expressions based on structural complexity in the unified CROHME 2014/2016/2019 dataset. RI denotes the relative improvement from DWAP to RLFN.

| $z\leftrightarrow2$ | $g\leftrightarrow9$ | $q\leftrightarrow9$ | $o\leftrightarrow0$ | $b\leftrightarrow6$ | Top-5 |
|---|---|---|---|---|---|
| -44% | -19% | 0% | -24% | 10% | -21% |

Table 6: Relative change of top 5 mis-recognition.

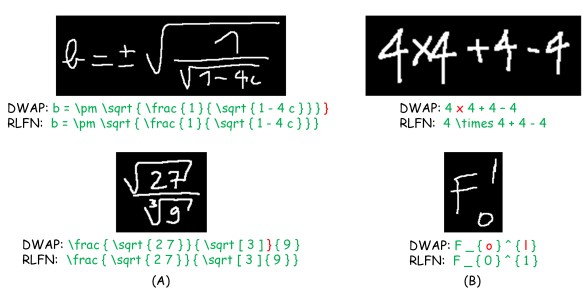

DWAP: b = \pm \sqrt { \frac { 1 } { \sqrt { 1 - 4 c } } } }
RLFN: b = \pm \sqrt { \frac { 1 } { \sqrt { 1 - 4 c } } }

DWAP: 4 x 4 + 4 - 4
RLFN: 4 \times 4 + 4 - 4

DWAP: \frac { \sqrt { 2 7 } } { \sqrt [ 3 ] } { 9 }
RLFN: \frac { \sqrt { 2 7 } } { \sqrt [ 3 ] { 9 } }

DWAP: F _ { o } ^ { l }
RLFN: F _ { 0 } ^ { 1 }

(A)     (B)

Figure 5: Cases with complex structure (A) and visual ambiguity (B) that RLFN outperforms our baseline.

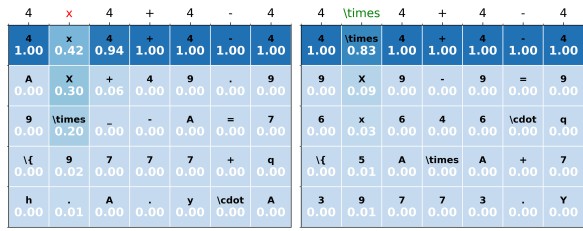

Figure 6: Visualization of top-5 probability

for the top 25% most complex expressions is nearly twice that of the improvement among all expressions. These observations indicate that RLFN outperforms our baseline especially in recognizing complex structures which might be because our RLFN can extract explicit structural information through LM, serving the similar role to tree decoders.

Regarding RLFN's performance in handling visual ambiguity, we conduct a replicated analysis same to the one described in Section 3.1. Specifically, we compare the frequency of top 5 number-alphabet pairs of mis-recognition with our baseline and study the difference, which is shown in Table 6. We observe that RLFN effectively reduces the occurrence of the top-5 mis-recognition by 21% compared to our baseline. This shows the capability of RLFN to reduce visual ambiguity between alphabets and numbers using contextual information provided by language modeling.

### 5.5 Case Study

As shown in Figure 5, we present two complex structure cases in group A, along with another two visual ambiguity cases in group B.

In group A, the baseline model recognizes an additional structure symbol '\}'in one case and misplaces '\}' to the wrong position in the other. In contrast, RLFN gets both expressions correctly. This is not a symbol recognition problem, which indicates that RLFN has learned the semantics of structure symbols and gained the ability of structure modeling.

In group B, cases possess visual ambiguity between variables and numbers. The baseline model relies solely on visual appearance and cannot distinguish visually resemble symbols. This is likely due to its lack of architecture to effectively utilize and comprehend contextual and structural information. RLFN, on the contrary, can correctly recognize '\times' with surrounding numbers and recognize '1' and '0' based on their superscript and subscript structural relations. This indicates that RLFN can complement each other's semantic and structural information when recognizing visually ambiguous symbols.

As depicted in Figure 6, we delve into a particular case about its top-5 probability. The probability derived from the baseline demonstrates visual ambiguity concerning the symbols 'x', 'X', and '\times' without understanding their semantics. After determining that this may be a multiplication structure, RLFN reduces the probability associated with 'x' and 'X', while recognizing the correct \times symbol with high confidence.

### 6 Conclusion

In this research, we have examined the unique language characteristics intrinsic to LaTeX mathematical expressions, with a keen focus on the minimal semantic unit and explicit structural symbols. Our investigation underscores that these characteristics give HMER systems the potential to obtain both se-

mantic and structural information through language models. We subsequently propose an innovative architecture that harmoniously integrates recognition and language features to yield corrected sequences. This framework eliminates the requirement to construct complex CFGs for resolving structural issues, and serves to ameliorate the challenge of visual ambiguities. This integrative approach offers fresh insights and promising theoretical groundwork for the development of HMER and related mathematical endeavors.

## Limitations

The limitations of our theoretical assessment warrant acknowledgment. In our analysis, we scrutinized the language characteristics of LaTeX mathematical expressions, drawing parallels between their expression mechanisms and those of the English language. This led us to posit that a LM adept at handling English semantics should, in theory, be equally proficient with LaTeX mathematical expressions. However, our methodology, rooted in inferential analogy, is weaker than directly analyzing how LMs handles LaTeX mathematical expressions and cannot be further extended, such as customizing a LM suitable for LaTeX mathematical expressions.

Our proposed model architecture is not devoid of certain limitations. The architectural design broadly follows a late fusion strategy, which, when contrasted with the early fusion approach seen in semantic modeling and modal fusion during the decoding phase of the recognition module, exhibits a lack of thorough information interaction. This shortfall is exemplified by our model's disregard for the prediction probability of the recognition sequence input to the language module, resulting in some information loss.

Besides, given the current state of the field, where most existing recognition models rely heavily on tree decoders and bidirectional training architectures, triplet data and reverse sequences are not suitable for language modeling. This limitation confines the range of selectable baseline models. Notwithstanding, one of our overarching goals in this endeavor is to maneuver around this intrinsic constraint that inherently stifles expansion.

Furthermore, as the formidable capabilities of LLM and LMM/MLLM are widely researched, some methods can even achieve an OCR-free understanding of text images. This casts doubt on the significance of excavating model architectures for specific tasks. However, is there a real need for the involvement of a general large model in a specific task? When the accuracy requirements are stringent, how do the upper limits of a general large model compare with that of a small model tailored for a specific task? Or is it the case that data is truly everything? These questions still require deep consideration.

## Acknowledgements

This work was supported in part by the National Natural Science Foundation of China under Grant No.62250057.

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
