# OpenReview forum: "Language Model is Suitable for Correction of Handwritten Mathematical Expressions Recognition"
_EMNLP/2023/Conference — EMNLP 2023 Main_

### Official Review · Reviewer_xgEb · 2023-07-28

**Soundness:** 3

**Excitement:**

4: Strong: This paper deepens the understanding of some phenomenon or lowers the barriers to an existing research direction.

**Missing References:**

-

**Paper Topic And Main Contributions:**

The paper adresses the problem of correcting the output of handwritten mathematical text recognition (which generates LaTeX output from images). The authors develop a model which does not rely on complicated structure focused methods but instead use an already introduced recognition module and a fine-tuned language model for LaTeX expressions (MathBERTa). A novel fusion model is introduced which is able to correct the output of the LaTeX sequences by utilizing structural and semantic information inherent in the language model and thus tries to solve the problem of visual ambiguity.

**Questions For The Authors:**

I have no questions.

**Reasons To Accept:**

The paper is well written, combines SOTA models in an innovative approach and improves partially the SOTA performance with respect to this special task. It avoids very specialized methods which have been proposed earlier and which are complicated to enhance to other situations. The new proposed network, its implementation and the optimization process are also well described.

**Reasons To Reject:**

Both components of the proposed fusion network are already known. The combination is novel but may be other possible combinations and architectures may perform even better but have not been discussed by the authors.

**Reproducibility:**

4: Could mostly reproduce the results, but there may be some variation because of sample variance or minor variations in their interpretation of the protocol or method.

**Reviewer Confidence:**

3: Pretty sure, but there's a chance I missed something. Although I have a good feel for this area in general, I did not carefully check the paper's details, e.g., the math, experimental design, or novelty.

**Typos Grammar Style And Presentation Improvements:**

-

---

> ### Author Rebuttal · Authors · 2023-08-29
>
> Thank you for your review.
>
> In the article, we only discussed the issues related to post-correction-like architectures and addressed the potential information loss of our current framework in the limitations section.
>
> More article space has been devoted to the characteristics of Latex mathematical expressions and their impact on LM.
>
> Undoubtedly, superior architectures exist, and we hope to further explore them in subsequent research.

---

### Official Review · Reviewer_Gsfw · 2023-08-04

**Soundness:** 4

**Excitement:**

4: Strong: This paper deepens the understanding of some phenomenon or lowers the barriers to an existing research direction.

**Paper Topic And Main Contributions:**

The paper focuses on Handwritten Mathematical Expression Recognition (HMER), converting images of handwritten math expressions into LaTeX sequences. Current HMER methods employ attention-based encoder-decoder architectures with tree decoders to capture expression structure. However, these methods rely on Context-Free Grammars (CFGs) and pregenerated triplet data, which, based on the authors claims, limit scalability and neglect visual ambiguity challenges.

The study examines unique language traits in LaTeX math expressions: explicit structural symbols and symbols (especially letters) as minimal units with context-dependent semantics. The authors suggest language models can provide both structural and semantic info, aiding HMER correction.

They propose a new architecture, Recognition and Language Fusion Network (RLFN), that integrates recognition and language features to generate corrected sequences.

Reported experimental results on CROHME datasets highlight RLFN's higher performance. This research's key contribution lies in the fusion of recognition and language features, which could advance HMER and lays groundwork for broader math-related handwriting recognition and language understanding.

**Questions For The Authors:**

A: The paper mentions that existing approaches in HMER are limited by CFGs and pregenerated triplet data. Could you provide more insights into how these limitations affect the scalability and adaptability of current methods, and how RLFN overcomes these issues?

B: Could you discuss the training process and data requirements for the Recognition and Language Fusion Network (RLFN)? How does the integration of recognition and language features affect the overall training complexity and computational demands?

C: In the limitations you mention the late fusion of the recognition module and the language module. Is there any proposal how to improve the interaction between both?

D: The final loss function is defined as the sum of the cross-entropy loss of the recognition prediction sequence probability and the the corrected prediction sequence probability with respect to the ground truth. Could you explain if you could observe a prioritization of one of them, or if a weighted with hyperparameters could help improve the  performance of your approach?

**Reasons To Accept:**

The paper is well structured, well written and it provides a clear explanation of the proposed approach. It also contains ablation studies, case and improvement studies to demonstrate the effectiveness of the RLFN method in HWER.

The tables, summaries and graphs are very helpful to understand the paper.

The model tackles a relevant challenge which is handwritten text recognition applied to mathematical expressions and LaTeX, achieving SOTA results in various COHRME datasets.

The potential of this approach could be beneficial for other areas, such as the standard handwritten text recognition or image captioning.

**Reasons To Reject:**

The data, code, method are neither made publicly available nor do the authors state that they plan to do so. So there is no possibility to check the claims the authors made about the effectiveness of their method.

More details about the hyperparameters would have been helpful to better understand the details.

**Reproducibility:**

5: Could easily reproduce the results.

**Reviewer Confidence:**

3: Pretty sure, but there's a chance I missed something. Although I have a good feel for this area in general, I did not carefully check the paper's details, e.g., the math, experimental design, or novelty.

**Typos Grammar Style And Presentation Improvements:**

l. 035: crossroad
l. 255: not a radical expression
l. 394: the recognition module
l. 549: wich might be because

---

> ### Author Rebuttal · Authors · 2023-08-29
>
> Thank you for your review.
>
>
>
> **Responses regarding code, data, question B, and question D:**
>
> Data, code, and weight files were packaged within the **Supplementary Materials** when the article was submitted. After downloading MathBERTa from huggingface, the provided weight files can be used to reproduce the results on the three test sets.
>
> In Section 5.1, we detailed critical parameters and specifics that have a direct bearing on the results. Notably, the GPU/batch size/epoch/time/optimizer settings are: one 3090/8/200/16 hours/Adadelta. The learning rate ascends from 0 to 1 during the first epoch and thereafter decays to 0, following the cosine schedules. No data augmentation for fair comparison. And all images within a batch are filled in the upper left corner of the canvas of the same size. Due to memory limitations, the canvas size does not exceed 1280 * 280, and any excess images will be discarded. Other hyperparameters can also be directly viewed in the code.
>
> For detailed discussions related to datasets and the application layer, please refer to our response to Reviewer jmUa. On the whole, the CROHME dataset is characterized by its limited training data and intricate structures, factors that may contribute to overfitting and weak generalization. Theoretically, furnishing supplementary information might be a viable solution. Our RLFN incorporates two means to provide supplementary information. On one hand, we introduce domain knowledge through MathBERTa, freezing its parameters and treating it as prior knowledge. On the other, we jointly optimize the recognition prediction sequence and the predicted sequence resulting from combining visual and linguistic features, hoping the model can strike a balance between these two sources of information.
>
> Practically speaking, the added training parameters come only from the fusion module. When the hidden layer dimension of the fusion module is set at 256, the increase in parameters is limited, increasing the model's size from ~4.7M to ~5.1M. The actual training time for a single epoch increases by about 20%, but the actual training difficulty decreases, making it easier to fit the training set. Regarding the training results, we noticed that the primary enhancement over the baseline is attributed to the joint optimization, directly enhancing the recognition module's predictive accuracy. The gains from the fusion module's prediction corrections vis-à-vis the recognition model's outputs are comparatively marginal.  We attempted to improve the effectiveness of 'correction' by using methods including weighting loss with a ratio of 1:2 and increasing the hidden layer dimension of the fusion module to increase model capacity. The experimental accuracy of several parameter combinations remained between 55%-57%, and we also failed to see a significant improvement in the fusion module prediction comparing to the recognition module prediction.
>
> Moreover, in ablation study, where we abandoned the fusion module and jointly optimized the recognition module predictions and language modules' ensuing predictions. We noticed a similar trend, where the primary enhancement is attributed to the joint optimization, directly enhancing the recognition module's predictive accuracy.
>
> In summary, regarding the source of model improvement, we can essentially conclude that semantic information plays a role similar to 'aware', akin to those methods based on tree decoders. It serves as auxiliary information or a means of regularization to jointly optimize the recognition model, rather than enhancing it through a two-stage approach of first recognizing then correcting. This aligns with the aim to avoid the hallucination issue of post-correction in mathematical formulas being often questioned.
>
> However, it's worth noting that the fusion module uses visual features as input. Removing the loss from the recognition module means it can't guide its classification head and serve as input to the language module. This makes it impossible to directly ablate the effects of joint optimization to validate our perspective.
>
>
>
> **Response for question A:**
>
> Firstly, current tree decoders and works related to CFG develop independently, making continuous exploration challenging. The reasons for this include that the complex CFG design prevents them from learning from each other themselves; The pre constructed triplet of data conversion makes it difficult to apply other methods to experiments due to non-standard data.
>
> Furthermore, they neglect challenges related to visual ambiguity. This stems both from the inherent nature of being context-free and their approach of translating structural symbols into relations and distinguishing them from general symbols, directly limiting their capacity for contextual modeling.
>
> Our RLFN, being based on a linear decoder, offers a certain degree of scalability. However, as discussed in our limitations section, there are certain challenges related to bidirectional decoders.
>
> Regarding the issue of visual ambiguity, it involves our viewpoint in Section 3.1:
>
> 1. The presence of an explicit symbolic structure allows the LM to directly model the structure.
> 2. Based on contextual semantics, visual ambiguities between categories can be corrected.
>
> Additionally, it's worth noting that structural symbols are not directly represented in images and are regarded as a special kind of symbol. This brings attention alignment issues into play during the decoding process. Both the current tree decoders and linear decoders don't differentiate in the context of images, which could be a primary reason for attention drift. However, this hypothesis has not yet been substantiated in our research.
>
>
>
> **Response for question C:**
>
> Late fusion methods, in comparison to post-correction, merge features instead of sequences, retaining some information from the predicted sequences of the recognition module. Similarly, during language modeling, while the predicted sequence is still fed as input, the predicted probabilities of the recognition sequence inputted into the language module are overlooked, leading to information loss.
>
> However, fusing information at every step of decoding can evidently prevent this loss of information, but it would incur a prohibitive computational cost.
>
> This might immediately bring to mind a series of Visual Language Processing (VLP) tasks, an area I haven't delved deeply into.

---

### Official Review · Reviewer_W2zF · 2023-08-08

**Soundness:** 4

**Excitement:**

4: Strong: This paper deepens the understanding of some phenomenon or lowers the barriers to an existing research direction.

**Paper Topic And Main Contributions:**

The paper presents a novel method for recognition of handwritten math expressions. The input is an image of such an expression, the output is latex code that generates such expression. The problems has been studied relatively lot recently, there is a series of CROHME datasets from three years and several models were proposed. The contribution of this paper is to incorporate a language model component in the model, which is a quite novel approach. The language model is adopted from another work and it is a RoBERTa based model fine-tuned on latex match expressions. The results are quite promising outperforming SOTA by a large margin.

**Reasons To Accept:**

A well written paper, interesting idea, well-done experiments, promising results, wide comparison with other methods.
I appreciate the attempt to motivate the use of LM by a series of analyses of the exiting datasets and common errors obtained by the current SOTA. The ablation study showing the actual contribution of the LM component and the fusion part.

**Reasons To Reject:**

The paper might be not so much relevant to the conference. The only relevant part is the use of the LM component and the motivation for using the LM component by comparing the Latex language to English.
I am not were convinced by the theoretical analysis in Section 3.1 which tries to explain why LM can help in the task, e.g. the argument that
because \sqrt[z]{x} is less common than x^{1/z} than the output should be rather \sqrt[2]{x}. I trust the results that LM can help the task, but is such situations are typical, some quantitative analysis should be done.

**Reproducibility:**

3: Could reproduce the results with some difficulty. The settings of parameters are underspecified or subjectively determined; the training/evaluation data are not widely available.

**Reviewer Confidence:**

3: Pretty sure, but there's a chance I missed something. Although I have a good feel for this area in general, I did not carefully check the paper's details, e.g., the math, experimental design, or novelty.

---

> ### Author Rebuttal · Authors · 2023-08-29
>
> Thank you for your review.
>
> I largely agree with the issues you pointed out regarding the theoretical analysis in Section 3.1. In the limitations section, we also mentioned this method based on analogical reasoning, comparing the expression mechanism of LaTeX with English language to demonstrate that the LM is effective for the HMER task is not sufficient.
>
> Intuitively, LMs might be applicable to most natural languages, but for mathematics, it isn’t as straightforward.
>
> There's a skepticism in the field, mirrored in studies focusing on post-correction for OCR. They argue that solely relying on an LM for correcting low-redundancy information, such as numerical data, might expose the process to biases, particularly from the probabilistic distortion inherent in models.
>
> Recent trends offer further evidence. Following the surge in LLM's popularity, tasks like mathQA, which bridge natural language to mathematical expressions, have shown progress. Yet, when directly grappling with mathematical formulas, LLM continues to face challenges, especially in areas like mathematical reasoning.
>
> We couldn't find an appropriate method to explain 'Why' from an overall LM perspective. Subsequently, we shifted our focus to identify what characteristics of mathematical formulas make LM suitable for HMER. Apart from the model, a significant portion of our paper, including Sections 3.1, 3.2, and 5.4, is dedicated to explanations from theoretical, statistical, and result improvement perspectives.
>
> Reflecting on the feedback, including issues highlighted by another reviewer concerning our introduction, we acknowledge that our paper's viewpoints might come across as somewhat muddled. We have identified two main arguments in Section 3.1 and plan to make corresponding adjustments to this section and the introduction in the future.
>
> 1. The presence of an explicit symbolic structure allows the LM to directly model the structure.
> 2. Based on contextual semantics, visual ambiguities between categories can be corrected.
>
> Corresponding content of Statistical Analysis in Section 3.2:
>
> 1. The explicit structural symbols do express their semantics.
> 2. The visual ambiguity between numbers and alphabets do exist.
>
> Corresponding content of Improvement Study in Section 5.4:
>
> 1. Complex structural expressions get more improvements.
> 2. Visual ambiguity mis-recognition has been effectively reduced.

---

### Official Review · Reviewer_jmUa · 2023-08-11

**Typos Grammar Style And Presentation Improvements:** None
**Soundness:** 3

**Excitement:**

3: Ambivalent: It has merits (e.g., it reports state-of-the-art results, the idea is nice), but there are key weaknesses (e.g., it describes incremental work), and it can significantly benefit from another round of revision. However, I won't object to accepting it if my co-reviewers champion it.

**Missing References:**

None

**Paper Topic And Main Contributions:**

The problem being addressed is recognition of mathematical expressions which are handwritten into latex statements. The paper addresses structural and visual ambiguities and builds an architecture RLFN to solve this. Though the architecture is an extension to DWAP however by making use of recognition and language module there is improvement in expression recognition than the baseline method.

**Questions For The Authors:**

1) The evaluation can be made more exhaustive by adding more sources other than CROHME. How will the designed system evaluate with low resolution scanned expressions or those written by less experienced people. How will it recognize wrong expressions. Will it autocorrect.
2) Comparison with frameworks/platforms like Google Lens, Socratic etc. which has made solving math problems easy for people would also be appreciated.

**Reasons To Accept:**

The paper helps in making use of language characteristics of LaTeX way of writing mathematical expressions and use this to resolve conflicts. Mathematical expressions can get complex and typical OCR which involved segmentation and recognition steps are not applicable for this. Hence the paper could be useful to scientific community and mathematicians.

**Reasons To Reject:**

The paper is not easy to understand. The language can be simplified and made more crisp. While i was reading the introduction it was not clear to me that what is the problem to be solved exactly and what is the novelty the paper is adding. Only after i read the entire paper i was able to understand the work entirely. Introduction should get the reader inquisitive and give an overall view of paper and make the reader interested in reading the paper entirely.

**Reproducibility:**

4: Could mostly reproduce the results, but there may be some variation because of sample variance or minor variations in their interpretation of the protocol or method.

**Reviewer Confidence:**

3: Pretty sure, but there's a chance I missed something. Although I have a good feel for this area in general, I did not carefully check the paper's details, e.g., the math, experimental design, or novelty.

---

> ### Author Rebuttal · Authors · 2023-08-29
>
> Thank you for your review.
>
>
>
> Firstly, I'd like to address the practical application concerns you've highlighted. While these may not directly tie into the central thrust of my paper, my response stems primarily from our experiences in trying to use HMER as the input for automated grading systems.
>
> To the best of my knowledge, current HMER systems still lag behind human performance. This includes the Google Lens you mentioned, as well as the frequently used Mathpix (which excels in recognizing individual formula images) and TAL (which excels in recognizing full answer sheet images) APIs. Despite them potentially being trained on vast amounts of data, their accuracy is still insufficient for applications demanding high precision, such as automated grading. Human intervention remains necessary in most scenarios.
>
> The CROHME dataset discussed in the paper is different. CROHME is artificially created and includes some exceedingly complex formulas that might rarely be encountered in real-world scenarios. The evaluation requires training on 8,835 images from the training set without any data augmentation. On one hand, this is to ensure a fair comparison; on the other hand, it measures the unique ability of HMER systems to handle complex structures, which typically represents the upper limit of the model.
>
> We've tested the Mathpix and TAL APIs on CROHME2014, and the accuracy rates for Ours RLFN/Mathpix/TAL are 0.5700/0.6187/0.2667 respectively. Regarding Google Lens, since we couldn't locate a dependable API, we evaluate the initial 100 test samples from CROHME2014 using its app, with accuracy rates of 63/100 for our RLFN and 61/100 for Google Lens. It's essential to note that this isn't a fair comparison. For such applications, on one hand, they probably have tenfold or even a hundredfold more training data, undoubtedly enhancing their capability. On the other hand, the diverse sources of their data introduce additional noise, potentially reducing performance.
>
> Outside of CROHME, real-world datasets, those containing low-resolution images or shadows, are seldom made public. We could only find an open dataset called HME100K, which covers segmented mathematical and chemical formulas. However, they don't provide data cleaning or augmentation methods. Compared to the reported 67% accuracy by the dataset provider, our RLFN achieved around 62% without data preprocessing. This discrepancy may be due to our exclusion of chemical formulas and the lack of additional data cleaning and augmentation.
>
> In our practice, we have labeled scanned images from the same source (with different writers). The performance metrics are as follows: RLFN training on HME100K/API(TAL)/RLFN fintune on ~1k data, 69%/ 78% / 80%. This indicates that providing data tailored to the scenario can result in a significant performance boost. However, for individual daily use, the previously mentioned APIs or software are undoubtedly more robust.
>
> Additionally, the HMER task doesn't specifically cater to less experienced people, and our model won't autonomously correct formulas devoid of image information. On the contrary, this is what we aim to avoid. The problem you've described seems more aligned with the requirements of an auxiliary input system, rather than the core objective of HMER – precise image recognition suited for high-precision tasks like automated grading.  Our choice of employing a fusion module over post-correction methods is driven by our intent to provide semantic context for visually ambiguous terms and to avoid dependency on semantic cues for high-confidence visual terms, thus preventing illusions or extreme corrections due to factors like low redundancy, such as the output x=1 being corrected to x=2.
>
>
>
>
> Lastly, we review the introduction section and the comments of another reviewer W2zF, I think it may be unclear when trying to introduce our viewpoints about what characteristics of mathematical formulas make LM suitable for HMER, resulting in confusion in paragraphs 4, 5, and 6 of the introduction.
>
> We have identified two main arguments in Section 3.1 and plan to make corresponding adjustments to this section and the introduction in the future.
>
> 1. The presence of an explicit symbolic structure allows the LM to directly model the structure.
> 2. Based on contextual semantics, visual ambiguities between categories can be corrected.
>
> Corresponding content of Statistical Analysis in Section 3.2:
>
> 1. The explicit structural symbols do express their semantics.
> 2. The visual ambiguity between numbers and alphabets do exist.
>
> Corresponding content of Improvement Study in Section 5.4:
>
> 1. Complex structural expressions get more improvements.
> 2. Visual ambiguity mis-recognition has been effectively reduced.

---

### Meta-Review · Area_Chair_ZbYw · 2023-09-18

**Recommendation:** 4

**Metareview:**

All four reviewers felt generally positively about the paper, citing the usefulness of the task, the interestingness of the idea, promising results, and wide comparison with other methods., and potential application to other tasks. One reviewer, jmUa, had a concern about presentation -- that the paper was only clear after reading it in entirely -- but other reviewers felt the paper was well-written. Reviewer W2zF had a concern about relevance of the task (parsing an image into latex code) to the conference, but seemed convinced by the author response and still felt positively about the paper. Overall this paper looks to make an interesting contribution on a well-motivated task, with thorough experiments and strong results.

---

### Decision · Program_Chairs · 2023-10-07

**Decision:**

Accept-Main

**Comment:**

All four reviewers felt generally positively about the paper, citing the usefulness of the task, the interestingness of the idea, promising results, and wide comparison with other methods., and potential application to other tasks. One reviewer, jmUa, had a concern about presentation -- that the paper was only clear after reading it in entirely -- but other reviewers felt the paper was well-written. Reviewer W2zF had a concern about relevance of the task (parsing an image into latex code) to the conference, but seemed convinced by the author response and still felt positively about the paper. Overall this paper looks to make an interesting contribution on a well-motivated task, with thorough experiments and strong results.